# In Vitro Systematic Drug Testing Reveals Carboplatin, Paclitaxel, and Alpelisib as a Potential Novel Combination Treatment for Adult Granulosa Cell Tumors

**DOI:** 10.3390/cancers13030368

**Published:** 2021-01-20

**Authors:** Joline Roze, Elena Sendino Garví, Ellen Stelloo, Christina Stangl, Ferdinando Sereno, Karen Duran, Jolijn Groeneweg, Sterre Paijens, Hans Nijman, Hannah van Meurs, Luc van Lonkhuijzen, Jurgen Piek, Christianne Lok, Geertruida Jonges, Petronella Witteveen, René Verheijen, Gijs van Haaften, Ronald Zweemer, Glen Monroe

**Affiliations:** 1Department of Gynaecological Oncology, UMC Utrecht Cancer Center, University Medical Center Utrecht, Utrecht University, 3584 CX Utrecht, The Netherlands; J.F.Roze@umcutrecht.nl (J.R.); J.W.Groeneweg-11@umcutrecht.nl (J.G.); rene.h.m.verheijen@gmail.com (R.V.); G.Monroe@umcutrecht.nl (G.M.); 2Department of Genetics, Center for Molecular Medicine, University Medical Center Utrecht, Oncode Institute, Utrecht University, 3584 CX Utrecht, The Netherlands; e.sendinogarvi@uu.nl (E.S.G.); E.Stelloo@umcutrecht.nl (E.S.); C.S.Stangl-2@umcutrecht.nl (C.S.); ferdinando.sereno@studenti.unipd.it (F.S.); K.J.Duran@umcutrecht.nl (K.D.); G.vanHaaften@umcutrecht.nl (G.v.H.); 3Department of Obstetrics and Gynaecology, University Medical Center Groningen, University of Groningen, 9713 GZ Groningen, The Netherlands; s.t.paijens@umcg.nl (S.P.); h.w.nijman@umcg.nl (H.N.); 4Department of Gynecological Oncology, Centre for Gynaecological Oncology Amsterdam, Amsterdam University Medical Center, 1105 AZ Amsterdam, The Netherlands; h.s.vanmeurs@amsterdamumc.nl (H.v.M.); l.r.vanlonkhuijzen@amsterdamumc.nl (L.v.L.); 5Department of Obstetrics and Gynaecology, Catharina Hospital, 5623 EJ Eindhoven, The Netherlands; jurgen.piek@catharinaziekenhuis.nl; 6Department of Gynaecological Oncology, Centre for Gynaecological Oncology Amsterdam, The Netherlands Cancer Institute, Antoni van Leeuwenhoek Hospital, 1066 CX Amsterdam, The Netherlands; c.lok@nki.nl; 7Department of Pathology, University Medical Center Utrecht, Utrecht University, 3584 CX Utrecht, The Netherlands; G.N.Jonges@umcutrecht.nl; 8Department of Medical Oncology, University Medical Center Utrecht, Utrecht University, 3584 CX Utrecht, The Netherlands; P.O.Witteveen@umcutrecht.nl

**Keywords:** granulosa cell tumors, ovarian cancer, drug screens, targeted treatment, alpelisib

## Abstract

**Simple Summary:**

Granulosa cell tumor treatment is challenging as there are few effective options besides surgery. In this study, we obtained tumor tissue from patients at surgery and cultured tumor cells in the laboratory. After sufficient expansion, we tested the effects of current treatments, such as chemotherapy and anti-hormonal treatment, and novel anti-cancer treatment options on cell survival. Results were generated within three weeks after tissue collection. We found that all drugs were ineffective when used as single treatments; however, some combinations were very effective. The PI3K protein inhibitor alpelisib was effective in combination with chemotherapy and achieved 50% cell death at assumed tolerable patient plasma concentrations. In conclusion, this study shows an approach to rapidly establish patient-derived cell lines for drug screens. The effectiveness of combined treatment with alpelisib and chemotherapy in granulosa cell tumors should be further investigated and may be a promising novel treatment option in patients with a granulosa cell tumor.

**Abstract:**

Adult granulosa cell tumors (AGCTs) arise from the estrogen-producing granulosa cells. Treatment of recurrence remains a clinical challenge, as systemic anti-hormonal treatment or chemotherapy is only effective in selected patients. We established a method to rapidly screen for drug responses in vitro using direct patient-derived cell lines in order to optimize treatment selection. The response to 11 monotherapies and 12 combination therapies, including chemotherapeutic, anti-hormonal, and targeted agents, were tested in 12 AGCT-patient-derived cell lines and an AGCT cell line (KGN). Drug screens were performed within 3 weeks after tissue collection by measurement of cell viability 72 h after drug application. The potential synergy of drug combinations was assessed. The human maximum drug plasma concentration (Cmax) and steady state (Css) thresholds obtained from available phase I/II clinical trials were used to predict potential toxicity in patients. Patient-derived AGCT cell lines demonstrated resistance to all monotherapies. All cell lines showed synergistic growth inhibition by combination treatment with carboplatin, paclitaxel, and alpelisib at a concentration needed to obtain 50% cell death (IC50) that are below the maximum achievable concentration in patients (IC50 < Cmax). We show that AGCT cell lines can be rapidly established and used for patient-specific in vitro drug testing, which may guide treatment decisions. Combination treatment with carboplatin, paclitaxel, and alpelisib was consistently effective in AGCT cell lines and should be further studied as a potential effective combination for AGCT treatment in patients.

## 1. Introduction

Adult granulosa cell tumors (AGCTs) represent a hormonally active, rare subtype of ovarian cancer arising from stromal granulosa cells. The disease has an incidence of 0.6–1.0 per 100,000 women worldwide [1,2,3,4]. Patients are suspected to have an AGCT when presenting with postmenopausal or irregular vaginal bleeding, abdominal pain, high plasma estrogen and inhibin levels, or ultrasound findings of a cystic or solid ovarian mass [3]. However, due to its rarity, AGCTs are often not preoperatively recognized, and therefore are mostly diagnosed at histopathological evaluation after surgery. Microscopically, AGCTs harbor granulosa cells with grooved nuclei, with or without other stromal cells. The classical Call–Exner bodies, areas of eosinophilic fluid surrounded by granulosa cells, are detected in 30–60% of cases [5,6,7]. Immunohistological staining AGCTs will be positive for inhibin and calretinin. When histopathology is inconclusive, *FOXL2* c.402C > G (C134W) mutation testing can be performed. This specific mutation is a hallmark of AGCTs and is present in 90–97% of patients [8,9].

Primary tumors are confined to one ovary (stage I disease) in 78–91% of patients, which are surgically removed [3]. However, recurrences occur in approximately 50% of patients and often require repeated debulking surgeries. Ultimately, 50–80% of patients with a recurrence will die of disease [9,10,11]. Due to its rarity, studies specifically designed for the treatment of AGCT are lacking, resulting in poor efficacy of treatment. Systemic treatment strategies for AGCTs are, therefore, based upon studies on more common ovarian cancer subtypes. First-line systemic treatment is currently derived from high-grade epithelial ovarian cancer treatment guidelines and consists of the combination of carboplatin and paclitaxel, although combination treatment with bleomycine, etoposide, and cisplatin (BEP) is also used. A prospective study showed that compared to BEP, carboplatin and paclitaxel had a progression-free survival of 27.7 months versus 19.7 months and demonstrated a more favorable side effect profile (NCT01042522). A retrospective analysis investigating the efficacy of a chemotherapeutic treatment in AGCT demonstrated a partial response in 11–25% and complete response in a further 11–26% of patients [12]. These studies included only 5–39 patients for evaluation with varying chemotherapy regimens, as large patient numbers in this rare tumor type are difficult to obtain.

Since AGCTs express hormone receptors, endocrine therapy was thought to be an effective treatment. Initially, the selective estrogen receptor modulator tamoxifen was utilized, and currently the aromatase inhibitor letrozole and the estrogen receptor antagonist fulvestrant are also given. A recent retrospective study showed that anti-estrogen treatment decreased the tumor load in 4 out of 22 AGCT patients (18%) [13]. The similar response rate and significantly fewer side effects of anti-hormonal treatment as compared to chemotherapy could warrant consideration of endocrine therapy as a first systemic treatment. However, the response rates remain low and benefits are only expected in a subset of patients. Moreover, several recent studies found no survival benefit after treatment with either chemotherapy or endocrine therapy in AGCT patients, emphasizing the need for novel treatment options [14,15,16,17,18,19].

Treatment of recurrent disease remains a clinical challenge, since effective systemic therapies are lacking. Clinical drug trials are difficult to perform in rare diseases, such as AGCT. Therefore, critical evaluation of current systemic treatment options is needed to identify potentially sensitive subgroups, as well as to identify promising novel targeted therapies. To date, only one study that performed a large-scale drug screen on AGCT-patient-derived cell lines has been published [20]. In this study, many individual drug compounds in seven AGCT cell lines and four drug combinations in the AGCT cell line KGN were tested. Paclitaxel combined with either the *SRC* tyrosine kinase inhibitor dasatinib or the mTOR inhibitor everolimus resulted in a synergistic response, and RNASeq established that the downstream targets of these drugs were abundantly expressed in AGCTs [20]. Previous studies have shown growth inhibition when these inhibitors are used as a monotherapy in KGN cells or in a granulosa cell tumor peritoneal carcinomatosis mouse model [21,22]. Further, in vitro drug testing may help to identify effective drug combinations and personalize treatment for AGCT patients.

Here, we demonstrate a method to rapidly establish AGCT cell lines from patient-derived tumors and to screen for drug responses. We confirmed the *FOXL2* c.402C > G mutation to verify the tumor cell origin in the established cell lines and tested current and novel systemic therapies, including drug combinations to assess potential synergy. We detect synergistic inhibitory effects on cell growth for the combination treatment of carboplatin and paclitaxel with the specific PIK3CA inhibitor alpelisib at concentrations that are clinically relevant for patients in vivo. We show that rapid, systematic, patient-specific AGCT drug screens are feasible and could be used to personalize treatment selection.

## 2. Results

### 2.1. Rapid Patient-Derived AGCT Cell Line Establishment and Systematic Drug Screening

A multicenter prospective study was performed to obtain fresh patient-derived tumor tissue immediately after surgery. We were able to establish short-term 2D cultures for 38 out of 48 tumors, resulting in a 79% success rate. The first established cell lines were used to optimize culture conditions and the drug screen experimental setup. Subsequently, the growth inhibitory effects of 11 drug compounds were investigated in vitro in 12 AGCT-patient-derived cell lines originating from five different patients. Drug screen results were obtained within an average of 3 weeks (median 20, range 12–30 days) after tissue collection. Of each cell line, three biological replicates containing two technical replicates each were used for the drug screens. Drug compounds were selected based on current AGCT treatment, hormone-positive breast cancer treatment, and novel targeted treatment in other cancers. The AGCT cell line establishment and drug screen setup is summarized in Figure 1. Sanger sequencing confirmed the heterozygous *FOXL2* c.402C > G mutation in all tumors and in 9 of the 12 cell lines (Table 1). The tumor origin was confirmed by CytoSNP-850K snp array in the remaining two *FOXL2* wild-type cell lines (GCPA096T1.II and GCPA113T1.I; Appendix A). Additionally, loss of the *FOXL2* wild-type allele was confirmed in GCPA113T1.II, harboring a hemizygous *FOXL2* c.402C > G mutation. We used the breast cancer cell line MCF-7 and neuroblastoma cell line SH-SY5Y as positive controls for hormone treatment and chemotherapy, respectively. The immortalized human granulosa cell line SVOG-3e was used to test selective sensitivity for targeted drugs.

### 2.2. Treatment with Chemotherapeutic, Anti-Hormonal, or Targeted Monotherapy Shows Inefficacy at Maximum Plasma Concentrations in All AGCT Cell Lines

The growth inhibitory effects of chemotherapeutic agents carboplatin and paclitaxel; the anti-hormonal drugs tamoxifen, letrozole, anastrozole, fulvestrant, and ulipristal; and the targeted drugs everolimus, alpelisib, dasatinib, and 6-THIO-2dG were tested as monotherapies in 12 direct patient-derived AGCT cell lines and the KGN cell line (Figure 2 and Appendix A). The targeted drugs everolimus and dasatinib were previously shown to result in AGCT cell line growth inhibition in combination with paclitaxel [20]. Alpelisib was chosen to target the PI3K/AKT/mTOR pathway, which has been identified in AGCT pathogenesis [20]. 6-THIO-2dG is a telomerase blocker that results in telomeric DNA damage in cells expressing telomerase, as *TERT* promoter mutations and *TERT* activation are common in AGCTs [23,24]. Ulipristal is a progesterone receptor blocker and was chosen to target progesterone, as a recent study identified a high progesterone receptor composite score associated with decreased recurrence-free and overall survival [25]. For these drugs, we obtained the values of the maximum drug concentration achieved in plasma (Cmax) and steady state drug concentration in plasma (Css) from phase I/II studies [26,27,28,29,30,31,32]. For 6-THIO-2dG, Cmax and Css values are not available. The control cell lines SH-SYS5 and MCF-7 were sensitive for the chemotherapeutic agents and anti-hormonal drugs, respectively(IC50 < Cmax and IC10 < Css; Appendix A; IC50: concentration needed to obtain 50% cell death; IC10: concentration needed to obtain 10% cell death). Anastrozole did not decrease cell viability at tolerable concentrations in MCF-7, in concordance with previous reports [33,34]. Furthermore, the SVOG-3e cell line showed potential sensitivity to alpelisib, as the IC50 was slightly above the Cmax (7.55 μM versus 6.9 μM; Appendix A). Dose response curves of the individual drugs demonstrated similar response profiles for all AGCT cell lines, including KGN. For all monotherapies, the IC50 exceeded the Cmax and Css values, suggesting that monotherapy drug concentrations required for 50% cell death are unlikely to be achieved in vivo (Figure 2; Appendix A). Moreover, for all anti-hormonal drugs, concentrations higher than the Css were needed to obtain 10% cell death (tamoxifen: Css = 0.11 μM, range IC10 = 6.04–15.03 μM, Figure 2).

### 2.3. Combination Treatment in KGN Shows Synergistic Effects and Allows for Drug Dose Reduction

In order to investigate combination therapies, we evaluated 12 drug combinations and tested for synergistic interactions in KGN cells. The drug dosages in combination were kept at a constant ratio throughout the experiment and based upon their monotherapy ranges (Table 2 and Table 3). KGN cells were most sensitive for the combinations of carboplatin, paclitaxel, and alpelisib (10:1:2 ratio); carboplatin, paclitaxel, and dasatinib (10:1:0.4 ratio); alpelisib with everolimus (2:1 ratio); and everolimus with tamoxifen (5:2 ratio) (Figure 3). Out of the twelve combinations tested, eight showed a certain degree of synergy (Combination Index (CI) < 1) (Appendix A). Therefore, significant dose reduction could be applied to the individual drugs when used in combination. Although multiple drug combinations demonstrated synergy, most of these combinations could not reach 50% cell death at maximum plasma concentrations for each individual drug (IC50 > Cmax or Css). However, the single combination that was effective at concentrations below the Cmax of each individual drug was carboplatin, paclitaxel, and alpelisib (Figure 3). This combination was efficacious and showed strong synergy (CI = 0.14). These findings in the KGN cell line suggest that carboplatin, paclitaxel, and alpelisib at a 10:1:2 constant ratio could be a safe, effective combination treatment.

### 2.4. The Combination of Carboplatin, Paclitaxel, and Alpelisib Is Also Consistently Effective in AGCT-Patient-Derived Cell Lines

To investigate the sensitivity of the patient-derived AGCT lines to combination therapies, we evaluated 12 drug combinations and tested them for synergistic interactions. Overall, the dose–response curves for the drug combinations differed among cell lines, indicating intra- and inter-patient drug sensitivity variation (Figure 4). In addition, the synergistic effects of similar drugs also varied among patients (Appendix A). Similar to KGN, the three most effective combinations in all cell lines included everolimus with alpelisib (at 1:2 ratio), carboplatin with paclitaxel and alpelisib (at 10:1:2 ratio), and everolimus with tamoxifen (at 5:2 ratio). However, the dose reduction of the combinations of either everolimus with alpelisib or everolimus with tamoxifen was not sufficient to enable the individual drugs to be below estimated tolerable plasma concentrations.

The combination of carboplatin, paclitaxel, and alpelisib showed similar drug response profiles in AGCTs (Figure 5) and consistent synergy in 11 of 12 (92%) tested cell lines (combination index values in the range of 0.11–0.88; Appendix A). Although carboplatin, paclitaxel, and alpelisib were all ineffective as monotherapies (median IC50 values of 343.32, 93.31, and 14.99 μM, which exceeded the corresponding Cmax values of 134.90, 4.27, and 6.92 μM, respectively), their synergistic interactions allowed for significant dose reduction when used in combination. This was the only combination with significant growth inhibition at values lower than the Cmax (in 10 of 12 cell lines, 83%), indicating potential anti-tumor activity with in vivo use. These findings corroborate the findings in the KGN cell line and suggest that a combination of carboplatin, paclitaxel, and alpelisib at a 10:1:2 constant ratio could be a safe, effective treatment for AGCT.

As alpelisib and everolimus specifically target the PI3K/Akt/mTOR pathway, we performed targeted next-generation sequencing to test for mutations in 64 cancer genes, including the PI3K pathway genes (Appendix A). We identified three variants in PI3K, although they were either intronic (*n* = 2) or synonymous (*n* = 1) and not predicted to affect gene function. No other pathogenic mutations in these targeted genes were detected, particularly not in the PI3K pathway.

### 2.5. FOXL2 Mutation Status Does Not Affect Response to Effective Drug Combinations

We compared the drug screen responses in three cell lines of patient GCPA113, which each have a different *FOXL2* mutation status (*FOXL2* wild-type, heterozygous mutant, and hemizygous mutant, respectively). *FOXL2* mutation status was ascertained to verify tumor cell origin of the cell lines and additional typing by snp array was performed in those lines in which the *FOXL2* mutation was not present (or present in hemizygous form). The hemizygous mutant cell line (GCPA113T1.II) was most resistant to alpelisib monotherapy (IC50 of 21.37 μM versus 9.59 and 11.53 μM; Appendix A). This cell line showed sensitivity to the three most effective combinations in general (everolimus with alpelisib, carboplatin with paclitaxel and alpelisib, everolimus with tamoxifen) and resistance to all other combination treatments (Figure 4). The combination of carboplatin, paclitaxel, and alpelisib was consistently effective in AGCT cell lines, regardless of their mutational pattern or tumor location.

## 3. Discussion

This is the first study to evaluate drug monotherapies and combination treatment in established short-term, patient-derived AGCT cell lines. With this approach, we were able to perform systematic drug screens in patient-derived AGCT cell lines and the AGCT model cell line KGN to test the efficacy, synergy, and potential human safety of current AGCT treatment, novel anti-cancer drugs, and their combinations within three weeks of tissue collection. We found that the combination of carboplatin, paclitaxel, and alpelisib was consistently effective and synergistic at individual drug concentrations deemed non-toxic for in vivo use in humans in 11 of 13 (85%) tested AGCT cell lines.

### 3.1. The PI3K Inhibitor Alpelisib

Alpelisib is an oral selective phosphatidylinositol 3-kinase (PI3K) p110α inhibitor, which has been identified as a novel promising targeted treatment in different cancer types. The PI3K signaling pathway is one of the most frequently dysregulated pathways in human cancers, which controls key cellular processes involved in cancer cell proliferation and survival [35]. Activation of the PI3K pathway frequently occurs via mutations in *PIK3CA*, which encodes the PI3K p110α catalytic subunit. Although a somatic *PIK3CA* mutation is only present in a small proportion of AGCTs, dysregulation of the PI3K/AKT pathway plays a major role in the pathogenesis of AGCTs [20,36,37]. Approximately 15–17% of AGCTs harbor a mutation in genes involved in this pathway [24,38].

### 3.2. PI3K Inhibition in Combination with Chemotherapy

Our study demonstrates the effectiveness of PI3K inhibition when combined with chemotherapy in AGCTs in vitro. As our study did not include tumors with a PI3K pathway mutation, the effect of this combination therapy may be even more pronounced in tumors with a *PIK3CA* variant. The combination of alpelisib with paclitaxel is currently being studied in breast cancer patients and advanced solid tumors [39,40]. In gastric cancer, alpelisib and paclitaxel demonstrated synergistic anti-proliferative and anti-migratory effects in *PIK3CA* mutant cells [41]. In a xenograft model utilizing gastric cancer cells, alpelisib and paclitaxel significantly increased apoptosis and tended to prolong the survival of tumor-bearing mice. Our study shows synergy for alpelisib in combination with chemotherapy, analogous to the previous observation in gastric cancer, and demonstrates consistent effectiveness in patient-derived AGCT cell lines.

### 3.3. Combined PI3K/mTOR Inhibition

The efficacy of alpelisib with everolimus was also promising in our analysis, albeit not at presumed tolerable plasma concentrations. Preliminary results of a phase Ib trial showed a manageable safety profile for this combination (NCT02077933). Trials testing alpelisib in combination with other targeted treatments such as monoclonal antibodies in breast cancer or MEK inhibitors in meningiomas are ongoing (NCT04208178, NCT03631953).

### 3.4. PI3K-mTOR Inhibition Combined with Anti-Hormonal Treatment

PI3K-mTOR pathway inhibition was also thought to be an effective strategy in combination with anti-hormonal treatment, since targeting the PI3K-mTOR pathway has the potential to restore sensitivity to estrogen receptor inhibition [42]. Therefore, studies are investigating this possibility using PI3K and mTOR inhibitors in combination with endocrine therapies. In hormone-receptor-positive breast cancer, adding alpelisib to the estrogen receptor blocker fulvestrant has been shown to prolong the progression-free survival in *PIK3CA* mutated tumors [43]. In addition, a randomized phase II trial in postmenopausal women with aromatase inhibitor resistant metastatic breast cancer showed that adding everolimus to tamoxifen increased the time to progression by 4.1 months and reduced the death risk by 55% [44]. These findings, together with our encouraging drug screen results for everolimus and tamoxifen, indicate that this strategy should be further studied in AGCTs.

### 3.5. Previous Drug Screen Studies

Previous drug screen studies on KGN cells have described in vitro efficacy of mTOR inhibitors combined with paclitaxel, tyrosine kinase inhibitors, and PPARγ activation combined with XIAP inhibition [20,21,45]. In our study, everolimus showed some degree of synergy with chemotherapy in 9 of the 13 cell lines, but did not belong to the most effective drug combinations. Furthermore, we confirmed the effectiveness of the tyrosine kinase inhibitor dasatinib combined with carboplatin and paclitaxel in KGN, although responses in our patient-derived AGCT cell lines varied widely. A previous study detected upregulation of the PI3K/AKT/mTOR pathway in AGCTs and found increased sensitivity to inhibitors of this pathway as compared to normal human granulosa–lutein (hGL) cells [20]. In contrast, we found increased sensitivity of the immortalized granulosa cell line (SVOG-3e) to alpelisib, as compared to the AGCT cell lines (IC50 7.55 μM versus median IC50 14.99 μM). This might be due to the fact that the PI3K/AKT/mTOR pathway also plays a crucial role in granulosa cell proliferation or that the immortalization process may have increased sensitivity to PI3K inhibition in this specific cell line [46]. The upregulation of this pathway in AGCTs, together with our drug screen results, suggest that PI3K-mTOR pathway inhibition may be an effective treatment strategy in AGCTs, particularly when utilizing synergistic drug combinations that allow for drug dose reduction.

### 3.6. The Limited Effects of Monotherapies

Our study confirmed the limited effects of current AGCT treatment strategies, including carboplatin, paclitaxel, and anti-hormonal treatments. All monotherapies were deemed ineffective, as the IC50 values, and IC10 values for the anti-hormonal drugs, exceeded the maximum and steady-state plasma concentrations. However, it may be difficult to simulate the effects of maintenance therapies such as anti-hormonal treatment with a single-dose application drug screen setup. Our study does not include AGCT cell lines sensitive to either chemotherapy or anti-hormonal treatment as a single therapy, due to the general resistance of AGCTs to these treatments. However, this drug screen approach could be used to identify individual patients that would respond to anti-hormonal treatment or chemotherapy in the future.

### 3.7. FOXL2 Mutation Status in Patient-Derived Cell Lines

In this study, a hemizygous *FOXL2* mutation was detected in one cell line and two cell lines were *FOXL2* wild-type. The copy number profiles of the *FOXL2* hemizygous mutant or *FOXL2* wild-type cell lines corresponded to the initial tumor, confirming tumor origin. Although GCPA113T1.I presented a distinct tumor copy number profile, deviations with the initial tumor were seen. This could be caused by sampling bias or cultivation of a subclone of the tumor. All cell lines showed similar sensitivity for the three most effective combinations (carboplatin with paclitaxel and alpelisib, and everolimus with tamoxifen or alpelisib), regardless of their *FOXL2* mutation status.

### 3.8. Estimating Efficacy of Drug Combinations

In order to test the efficacy of drug combinations, we assessed the IC50 concentrations on a linear distribution for each cell line and drug individually, and assessed drug ratios by their synergy. Hereby, we were able to test drug combinations in different concentrations while maintaining a constant ratio. In general, in vitro IC50 concentrations are usually an accurate reflection of a drug’s efficacy in vivo. However, these values are a proxy, as IC50s may occasionally not reflect the achieved cytotoxicity in vivo [47]. A limitation of this study is that Cmax concentrations assessed for monotherapy use were applied, whereas ideally Cmax concentrations should also be established for each drug in combination. However, Cmax values for drug combinations are usually not available, as large clinical trials are required to test specific drug combination dosages and ratios. Moreover, for some drugs used in vivo, dosages are calculated based on the surface area (mg/mm^2^), which is not possible in vitro. However, drug concentrations for future in vivo use can potentially be derived from current clinical trials testing these drug combinations in other cancer types.

### 3.9. A Robust Drug Screen Model

Finding an effective, targeted treatment for AGCTs has been the subject of research over the past decade. As clinical trials for rare tumors will take many years, robust pre-clinical models are needed to screen for potentially effective treatment strategies. This study illustrates an approach to establish patient-specific AGCT cell cultures directly from tumors at high success rates and to rapidly screen AGCTs for potentially effective treatments. Although we aimed to find personalized treatment options for some of the patients, similar response profiles were seen for the three most effective combinations, indicating that in general AGCTs could be sensitive to these combinations. When applying this method to an increased number of patient-derived AGCT cell lines or treatment options, we may find specific differential drug responses amongst patients. Future studies utilizing molecular and cellular markers confirming the specific pharmacodynamic response to these and other promising novel monotherapies and combinations may be necessary prior to clinical applicability. Furthermore, preclinical studies in PDX models can be used to replicate and validate the findings presented within this study in a more complex environment containing the appropriate hormonal milieu, vasculature, immune response, and the tumor microenvironment, as has been performed previously in granulosa cell tumors [48]. The established drug screen model can constantly be adapted, and therefore can easily be used for novel drug combination testing and personalized drug selection in the future. The clinical use and effectiveness of these drugs in other cancer types will enable fast translation to the clinic.

## 4. Materials and Methods

### 4.1. Patient Recruitment and Tumor Tissue Acquisition

We conducted a national prospective study to obtain fresh patient-derived tumor tissue. Patients were included in five hospitals between 2018 and 2020. Ethical approval was obtained (UMCU METC 17-868) and all participants provided written informed consent. Tissue samples were obtained directly in the operating room or the adjacent pathology department and placed in advanced DMEM/F12 (Thermo Fischer Scientific, Waltham, MA, USA), 1% glutamax (Thermo Fischer Scientific, Waltham, MA, USA), 10 mM HEPES Buffer (Thermo Fischer Scientific, Waltham, MA, USA), 100 U/mL penicillin, and 100 ug/mL streptomycin (Sigma-Aldrich, St. Louis, MO, USA) (resulting medium: ADMEM+++). Samples were transported at room temperature and processed within 4 h.

### 4.2. Tumor Tissue Processing and 2D Cell Line Establishment

Tumor tissue samples were mechanically sheared with scalpels to single cells and very small tissue particles. Cells and tissue samples were collected by flushing the petri dish with ADMEM+++. Upon collection, the collection tube was inverted and large particulates were allowed to settle for 30 s, while the upper layer of the cell suspension was collected for further processing. The upper layer of the cell suspension was centrifuged for 5 min at 800 rpm and the resulting supernatant was removed via pipette and discarded. Red blood cell (RBC) lysis buffer (Sigma-Aldrich, St. Louis, MO, USA) was added as needed (0.5–2 mL) to the cell pellet, mixed for 3 min, and then centrifuged for 5 min at 800 rpm. Once single cells were isolated and RBC was removed, single cells were seeded onto adherent cell culture plates in ADMEM+++ supplemented with 10% fetal calf serum (Thermo Fischer Scientific; FCS, Waltham, MA, USA) and stored at 37 °C/5% CO_2_ in humidified incubators. The medium was replaced every 3–4 days and cell lines were passaged upon reaching ~80% confluency (approximately every 1–2 weeks). Passaging was performed by removal of the medium, washing with PBS, and detachment from the plate with TryplE (1X, Thermo Fischer Scientific, Waltham, MA, USA). All utilized cell lines were mycoplasma-negative. The presence of the AGCT-specific *FOXL2* c.402C > G mutation was verified with the primers 5′-CCGGCATCTACCAGTACATCA and 5′-GGAAGGGCCTCTTCATGC using the Acccuprime GC-rich polymerase: 5′ at 95 °C followed by (30″ at 95 °C, 30″ at 56 °C, 30″ at 72 °C) for 35 cycles and ended by 5′ at 72 °C. In case of a homozygous *FOXL2* c.402C > G mutation or *FOXL2* wild-type genotype, a CytoSNP-850k array was performed on the cell line and corresponding tumor DNA to verify the tumor origin and identify loss of heterozygosity.

### 4.3. Targeted Pathway Sequencing

DNA from the fresh-frozen pulverized tumor tissue was isolated using a DNEasy Blood and Tissue Kit (Qiagen, Venlo, The Netherlands). Targeted next-generation sequencing with a mean coverage of 500X coverage was performed on the Ion Torrent S5 system (ThermoFischer Scientific) using a custom next-generation sequencing (NGS) panel based on the Ion Ampliseq™ Cancer Hotspot Panel targeting mutational hotspots of 64 cancer-related genes. Variants with an allele frequency of at least 5% were reported. Variant call files were generated and analysis was performed using Alissa (Agilent Technologies Alissa Interpret v5.1.7).

### 4.4. Control Cell Models

Control cell lines were used as positive controls (MCF-7, SH-SY5Y) and to test differential response to treatment (SVOG-3e). MCF-7, an estrogen- and progesterone-receptor-positive human breast cancer cell line, was used as positive control for the anti-hormonal therapies [49]. Additionally, SH-SY5Y, a human neuroblastoma cell line, was used as positive control for chemotherapeutic agents [50]. KGN, a human AGCT cell line derived from a 63-year-old patient heterozygous for *FOXL2* c.402C > G, was used as an additional AGCT cell line to evaluate potential AGCT response [51]. Finally, the immortalized human granulosa cell line SVOG-3e [52] was used to assess selective drug response in AGCT versus healthy granulosa cells. SH-SY5Y and MCF-7 were cultured in DMEM/F12 (Thermo Fischer Scientific) + 10% FBS and 1% Pen/Strep and RPMI 1640 (Thermo Fischer Scientific, Waltham, MA, USA) + 10% FBS and 1% Pen/Strep, respectively. KGN was cultured in conditions identical to AGCT-patient-derived cell lines. SVOG-3e was cultured in a 1:1 ratio of Media 199 and Media 105 (Sigma-Aldrich, St. Louis, MO, USA) with 5% FBS and 1% Pen/Strep.

### 4.5. AGCT Viability Assessment in Response to Monotherapy and Combination Treatment

Cell lines were seeded at a density of 500 cells/well in 40 uL in 384-well plates (Corning, NY, USA) with the Multidrop™ cell dispenser (Thermo Fischer Scientific, Waltham, MA, USA) and sealed with Breathe-Easy^®^ membrane (Sigma-Aldrich, St. Louis, MO, USA) to prevent evaporation. The SVOG-3e line was seeded at 1000 cells/well due to contact-dependent growth requirements. For each cell line, three biological repeats and two technical repeats each were performed in order to ensure the validity of the drug screens. Cells were grown for 24 h prior to drug dispensing in 37 °C, 5% CO_2_, and 95% relative humidity. Test substances were dissolved in 100% DMSO or MQ + 0.01% Tween-20 in the case of carboplatin, as DMSO inactivates platinum-based treatments (Table 2) [53]. Subsequently, 11 drugs and 12 drug combinations were dispensed and all wells (excluding blanks and non-DMSO control wells) were normalized to 1% DMSO using the HP Tecan D300e Digital Dispenser (Hewlett-Packard, Palo Alto, CA, USA). All drugs were applied in linear concentration ranges (Table 2), including 16 data points to enable combination drug testing at constant ratios (Table 3). Next, the plates were covered with the Breathe-Easy membrane and incubated. After 72 h, the Multidrop dispensed 15 uL per well of Cell Titer Glo 2.0 (Thermo Fischer Scientific, Waltham, MA, USA) and plates were incubated in the dark for 10 min. After incubation, luminescence excitation was assessed in the Spectramax reader (Molecular Devices, San Jose, CA, USA) (top read, luminescence excitation at 500 nm) and raw data were processed in Microsoft Excel to subtract background luminescence. The Z-factor, a mathematical method used to quantify the quality of a drug screen based on the calculation of the separation band between the values of the positive and negative controls, was utilized to assess the quality of drug screens [54]. Screens with a Z-factor value of >0.5 were used for further analysis. To optimize the systematic AGCT drug screens, we used 29 AGCT cell lines to assess the individual drug concentration ranges and drug ratios for combination drug testing. These cell lines were not included in the final drug screens, since all cells were utilized.

### 4.6. Efficacy and Safety of Monotherapies

Overall cell survival (in %) was plotted against the drug concentration range of each drug (in log10[μM]) using GraphPad Prism 8.3.1. Values that exceeded 100% survival (within +20% biological deviation) were normalized to 100% survival. Similarly, values below 0% cell survival (within −20% biological deviation) were normalized to 0% survival. Values exceeding this deviation were excluded. Dose–response curves were obtained using the model “log(inhibitor) vs. variable slope (normalized response)” to fit the dose–response curves and R^2^ values were used to indicate how well the model fit the data (Appendix A). We interpolated cell survival to 50% to obtain the corresponding IC50 values for each drug and their 95% confidence intervals (inhibitory concentration—IC). The maximum plasma concentration (Cmax) and the plasma concentration at steady state (Css) were used as toxicity thresholds. The Cmax was utilized for drugs in which short, high-dosage time courses are required, while the Css was used for maintenance therapy regimes such as anti-hormonal treatment. Both the Cmax and Css are estimated during phase I and phase II clinical trials and provide an approximated threshold of drug toxicity in humans [26,27,28,29,30,31,32]. A drug was deemed effective when 50% cell death (IC50) was achieved at a concentration below the maximum plasma concentration (Cmax) or steady-state plasma concentration (Css) in vivo. The IC50 values were calculated based on the Chou–Talalay method using Compusyn software [55]. For anti-hormonal treatment, mostly provided as maintenance therapy, we used IC10 values in addition to the IC50.

### 4.7. Combination Treatment for AGCTs

In addition to the 11 monotherapies, 12 drug combinations were tested (Table 3) simultaneously with the individual drug treatments in the same 384-well plate. To ensure that the contribution of each drug in combination was similar for all data points, drugs in combination were applied in a constant ratio. Synergy was evaluated using the mass action law, a systematic analysis assessing dose–response dynamics in a cost-effective manner [56,57]. The combination index (CI) was used to indicate antagonistic (CI > 1) additive (CI = 1) and synergistic (CI < 1) effects [55].

Similar to monotherapies, the safety for in vivo use of drug combinations as assessed by comparison to Cmax or Css values of the individual drugs (Appendix A). The dose reduction index (DRI) value for each drug in combination was obtained from the mass action law [56,57], indicating the dosage a drug can be reduced in combination to obtain the equivalent amount of cell death in monotherapy. A DRI value of >1 allows dose reduction. Effect calculations and data visualization were performed using both GraphPad Prism 8.3.1 and Compusyn software [58]. Finally, the IC50 values of the drugs in combination were calculated and compared to the Cmax and Css values. A drug combination was deesmed effective when 50% cell death (IC50) for all individual compounds was achieved at concentrations below Cmax or Css.

## 5. Conclusions

This study shows that rapid, systematic, patient-specific AGCT drug screens are feasible and can be used to test individual responses to existing monotherapies and novel combinations. The findings from a set of 13 cell lines demonstrated synergistic growth inhibition of the PI3K inhibitor alpelisib combined with the current first-line chemotherapeutic agents carboplatin and paclitaxel at potentially tolerable concentrations in vivo. Therefore, alpelisib, carboplatin, and paclitaxel may be a promising novel combination for treatment of recurrent AGCTs.

## Figures and Tables

**Figure 1 cancers-13-00368-f001:**
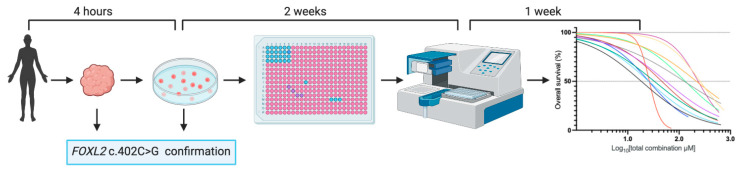
AGCT cell line establishment and drug screen work flow. Tumor samples were processed within 4 h after collection. Confirmation of the *FOXL2* c.402C > G mutational status of the tumor and cell line was performed by Sanger sequencing. Upon sufficient expansion, cells were seeded in a 384 well plate and drugs were applied 24 h after cell seeding. The response to treatment was analyzed 72 h after drug application by measuring cell viability.

**Figure 2 cancers-13-00368-f002:**
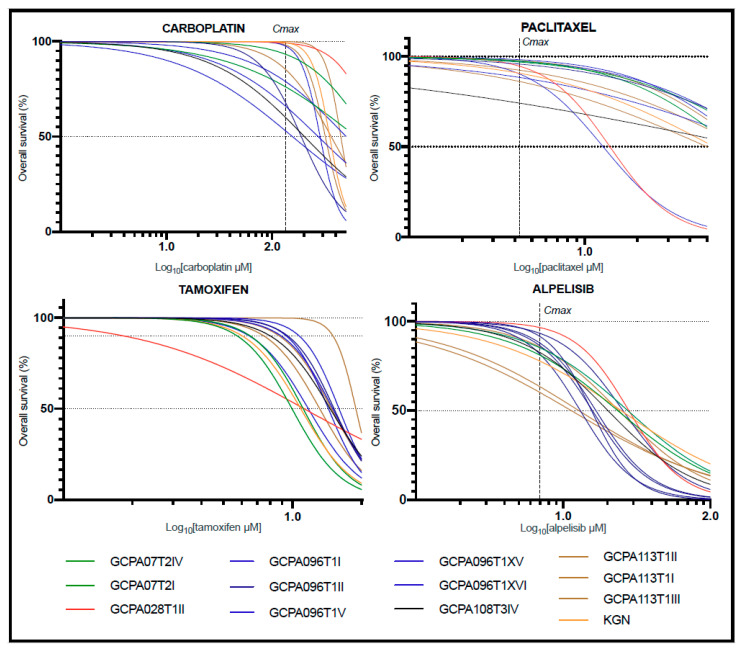
Representative drug screen results of four out of eleven tested monotherapies. For all monotherapies, the IC50 exceeded the Cmax and Css values. This suggests that monotherapy drug concentrations needed to achieve 50% cell death could not be achieved in vivo. Since the Log_10_ (Css) of tamoxifen is below 0, the Css is not displayed. Additionally, the concentration tamoxifen needed to achieve 10% cell death (IC10) also exceeded the Css. Each curve represents the average response of three biological replicates and two technical replicates. The Z-factor was ≥0.8 for all drug screens. IC50: concentration needed to obtain 50% cell death; IC10: concentration needed to obtain 10% cell death; Cmax: maximum drug concentration achieved in plasma; Css: steady state drug concentration in plasma.

**Figure 3 cancers-13-00368-f003:**
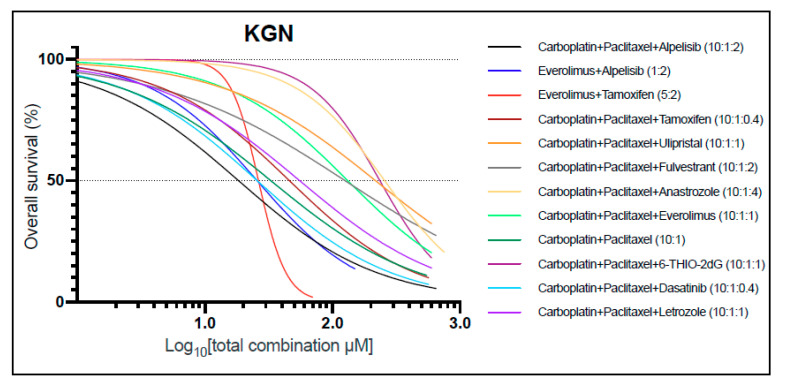
KGN responses to drug combination treatments. Dose–response curves show differential sensitivity to drug combinations. Drug combinations were applied at a constant ratio to ensure a constant contribution of each drug in combination for all data points. Each curve represents the average response of three biological replicates and two technical replicates. Z-factor: 0.91.

**Figure 4 cancers-13-00368-f004:**
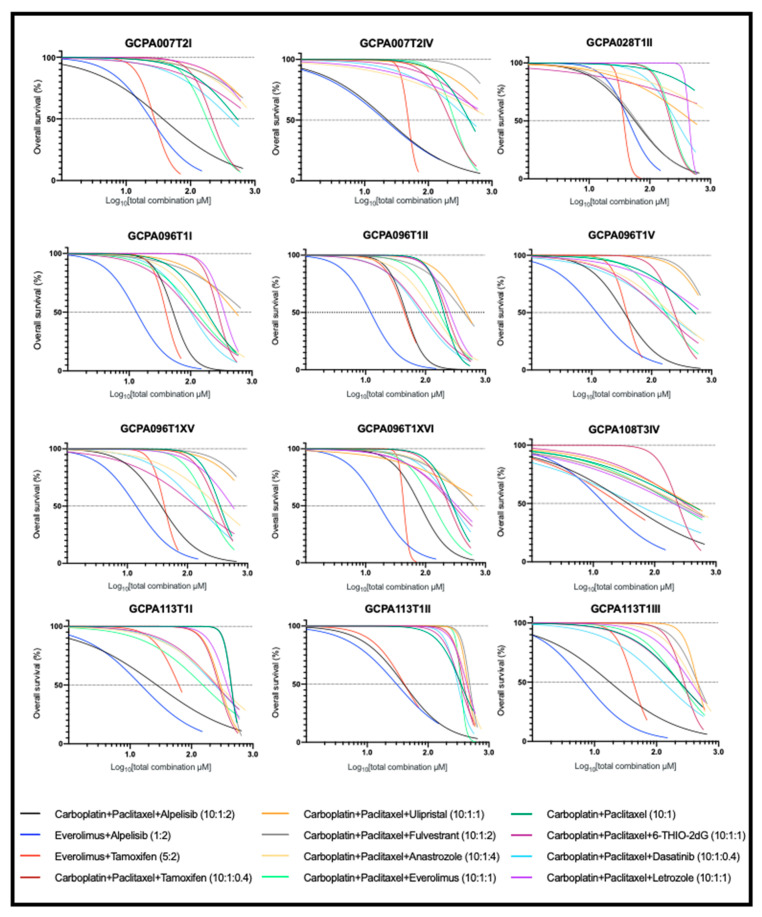
Patient-derived AGCT cell line response to drug combination treatment. Dose–response curves show differential sensitivity to drug combinations. Drug combinations were applied at a constant ratio to ensure a constant contribution of each drug in combination for all data points. Each curve represents the average response of three biological replicates and two technical replicates. The Z-factor was ≥0.8 for all drug screens.

**Figure 5 cancers-13-00368-f005:**
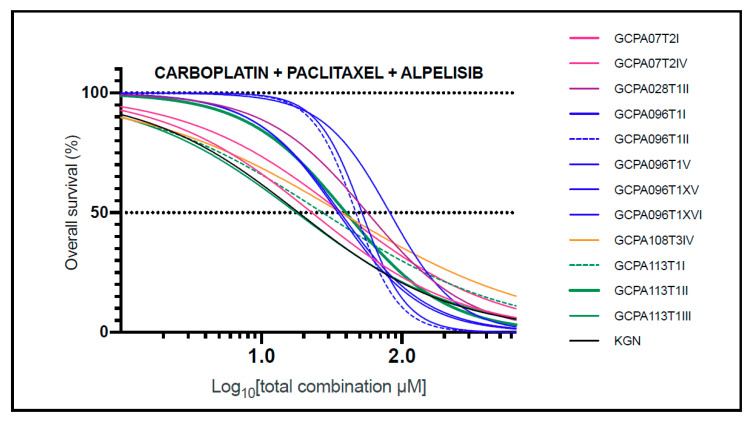
AGCT cell line response to combination treatment with carboplatin, paclitaxel, and alpelisib. Pearson’s correlation coefficient (r) was used to evaluate the association between two response profiles and ranged from 0.84 to 0.99 (mean: 0.95). In 10 of 12 cell lines, the IC50 was lower than the Cmax, indicating effectiveness at non-toxic levels in vivo. Each curve represents the average response of three biological replicates and two technical replicates. The Z-factor was ≥0.8 for all drug screens. IC50: concentration needed to obtain 50% cell death; Cmax: maximum drug concentration achieved in plasma.

**Table 1 cancers-13-00368-t001:** Patient-derived cell line characteristics.

Cell Line	Tumor Origin	Tumor Type	Previous Systemic Treatment	Tumo *FOXL2* c.402C > G Mutational Status	Cell Line *FOXL2* c.402C > G Mutational Status
**Direct Patient-Derived Cell Lines**					
**GCPA007**	AGCT	Recurrence	Radiotherapy, chemotherapy ^1^		
T2.I				+/−	+/−
T2.IV				+/−	+/−
**GCPA028**	AGCT	Recurrence	No		
T1.II				+/−	+/−
**GCPA096**	AGCT	Recurrence	No		
T1.I				+/−	+/−
T1.II				+/−	−/− ^3^
T1.V				+/−	+/−
T1.XV				+/−	+/−
T1.XVI				+/−	+/−
**GCPA108**	AGCT	Recurrence	Anti-hormonal treatment, chemotherapy, RFA ^2^		
T3.IV				+/−	+/−
**GCPA113**	AGCT	Recurrence	No		
T1.I				+/−	−/− ^3^
T1.II				+/−	+/+ ^3^
T1.III				+/−	+/−
**Control cell lines**					
**KGN**	AGCT	Primary	No	+/−	+/−
**SVOG-3e**	Granulosa cells	N/A	No	N/A	N/A
**MCF-7**	Breast cancer	Recurrence	Radiotherapy, anti-hormonal treatment	N/A	N/A
**SH-SY5Y**	Neuroblastoma	Recurrence	Radiotherapy, chemotherapy	N/A	N/A

Sample IDs: first number indicates time point, second number indicates location (e.g., T2.I is the first location of the second time point); N/A: not applicable; AGCT: adult granulosa cell tumor.^1^ Four cycles of bleomycine, etoposide, and cisplatin. ^2^ Anti-hormonal treatment: aromatase inhibitor anastrozole, progestogen megestrol, and selective estrogen receptor modulator tamoxifen, four cycles of carboplatin and paclitaxel, radiofrequency ablation (RFA) on liver metastases. ^3^ In these cell lines, tumor origin was confirmed by CytoSNP-850K snp array.

**Table 2 cancers-13-00368-t002:** Drug compounds.

Drug	Mechanism	Concentration Range (μM)	Solvent
Carboplatin	Intra- and inter-strand cross-linkage of DNA	500–0	MQ + 0.01% Tween
Paclitaxel	Microtubule stabilizer, induces mitotic arrest	50–0	DMSO
Tamoxifen	Estrogen receptor blocker	20–0	DMSO
Letrozole	Aromatase inhibitor	50–0	DMSO
Fulvestrant	Estrogen receptor blocker	100–0	DMSO
Ulipristal	Progesterone receptor blocker	50–0	DMSO
Anastrozole	Aromatase inhibitor	200–0	DMSO
Everolimus	mTOR inhibitor	50–0	DMSO
Alpelisib	PI3K inhibitor	100–0	DMSO
Dasatinib	Tyrosin kinase inhibitor	20–0	DMSO
6-THIO-2dG	Telomerase blocker	50–0	DMSO

All drug compounds were obtained from Sigma-Aldrich.

**Table 3 cancers-13-00368-t003:** Drug combinations and ratios.

Drug Combination	Combination Ratio
Carboplatin + Paclitaxel	10:1
Carboplatin + Paclitaxel + Tamoxifen	10:1:0.4
Carboplatin + Paclitaxel + Letrozole	10:1:1
Carboplatin + Paclitaxel + Fulvestrant	10:1:2
Carboplatin + Paclitaxel + Ulipristal	10:1:1
Carboplatin + Paclitaxel + Anastrozole	10:1:4
Carboplatin + Paclitaxel + Everolimus	10:1:1
Carboplatin + Paclitaxel + Alpelisib	10:1:2
Carboplatin + Paclitaxel + Dasatinib	10:1:0.4
Carboplatin + Paclitaxel + 6-THIO-2dG	10:1:1
Everolimus + Tamoxifen	5:2
Everolimus + Alpelisib	1:2

## Data Availability

The data presented in this study are openly available in FigShare at https://doi.org/10.6084/m9.figshare.c.5275025.

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
