# Peer review of "In Vitro Systematic Drug Testing Reveals Carboplatin, Paclitaxel, and Alpelisib as a Potential Novel Combination Treatment for Adult Granulosa Cell Tumors"

_cancers, 2021, doi:10.3390/cancers13030368_

Round 1

Reviewer 1 Report

The authors developed 12 patient derived cell lines from women with recurrent granulosa cell tumors (a rare tumor), which they used to screen both monotherapy and combination therapies. They found synergism of the combination of carboplatin, paclitaxel and alpelisib. This combination should be considered for a clinical trial. The manuscript is overall well written and easy to read. I do have some questions, comments for the authors.

  1. GOG 264- which compared carbo, taxol to BEP was presented in poster form at IGCS by Dr. Jubliee Brown- these results should be referenced in the background. Median PFS for carbo, taxol was 27.7 mo v 19.7 mo for BEP.
  2. Why were additional chemotherapies not tested?
  3. I would not expect some of these therapies to be cytotoxic- especially in cell lines- for example tamoxifen, AI, fuvestrant, ulipristal- one would likely need hormonal mileu to test their effectiveness. Cell lines not best way to assess efficacy of these treatments for granulosa cell tumors.
  4. I think some explanation as to why you chose to test some of these drugs is warranted- for example 6-TH10-2dG and dasatinib.

Reviewer 2 Report

    1. Review of Screening for Personalized Treatment of Granulosa 2 Cell Tumors: Synergistic Effect of Carboplatin, 3 Paclitaxel and Alpelisib 4
    2. Authors Joline Roze 1,†, Elena Sendino Garví 2,†, Ellen Stelloo 2, Christina Stangl 2, Ferdinando Sereno 2, 5 Karen Duran 2, Jolijn Groeneweg 1, Sterre Paijens 3, Hans Nijman 3, Hannah van Meurs 4, Luc van 6 Lonkhuijzen 4, Jurgen Piek 5, Christianne Lok 6, Geertruida Jonges 7, Petronella Witteveen 8, René 7 Verheijen 1, Gijs van Haaften 2, Ronald Zweemer 1 and Glen Monroe 1,*
    3.  
  • Brief Summary of the research conducted and summarized in this manuscript. 
  1. Adult granulosa cell tumors are a rare form of cancer and the low numbers of patients available preclude traditional pathways of clinical trials to deliver timely improvements in cancer treatments to patients. The submitting authors investigated the sensitivity of adult granulosa cell tumors (AGCTs) to anti-hormonal treatment or chemotherapy treatments applied to patient-derived cells. The investigators anticipate that drug responses in vitro could rapidly lead to optimized treatment selection for these patients. The response to 11 monotherapies and 12 combination therapies, were tested in 12 AGCT patient-derived cell lines and an AGCT cell line (KGN) relative to breast cancer cell line MCF-7 and neuroblastoma cell line SH-SY5Y. The investigators cited maximum plasma concentration thresholds (Cmax/Css), obtained from available phase I/II clinical trials, as achievable and well-tolerated concentrations of drug substances for the 1 monotherapies used in this work and found AGCT cell lines were resistant to all monotherapies. In subsequent experiments, the authors suggest that all cell lines showed synergistic growth inhibition by combination treatment with carboplatin, paclitaxel and alpelisib that are below the maximum achievable concentration in patients (IC50<Cmax).
  2. Originality/Novelty: The investigators have taken an interesting approach to evaluate treatment efficacies of various approved clinical products as monotherapy or combination therapy for adults granulosa cell tumors. It would be anticipated that robust responses in vitro obtained from this report would lead to recommendations to test 1 or 2 combination therapies in patients with AGCT as either first-line or second-line therapy for these ovarian-derived tumors. The methods used for assessing treatment efficacy of monotherapy or combination therapy are not original or novel nor are the results fully convincing that the recommended 3-drug combination therapy provides the best path forward for treatment of patients bearing AGCTs.
  3. Interest to the Readers: Although AGCTs have a much lower incidence than other forms of ovarian cancer, there is nonetheless strong interest to determine if in vitro drug sensitivity results can be predictive of in vivo clinical responses for these tumors, or generally for many rare forms of tumors. Convincing results of on-target engagement and demonstration of equivalent drug exposure as montherapy or combination therapy would significantly increase the confidence of readers that the treatment effects in cells can be predictive of clinical responses. The 2-drug combination of everolimus+apelisib provided consistently better control of growth of AGCTs across all genotypes of AGCTs than the 3-drug combination of carboplatin+paclitaxel+apelisib. In a clinical trial setting a 2-drug combination therapy would be much simpler to manage than a 3-drug combination. In this context, an important question is whether the addition of chemotherapy agent improves outcome of the double signaling pathway inhibitors (PI3K/mTOR) or whether there is no additional benefit of chemotherapy agent.
  4. Overall Merit: Moderate
  5. The manuscript is organized according to the requested format for this journal. The Supplementary files were not found in the online portal for reviewers.
  6. BROAD COMMENTS
  7. Significance:
  8. The author’s review of the literature identifies previous efforts by other authors (2017) that would be useful to review in greater detail in introduction to describe prior work on establishing sensitivity of the cells to dasatanib and paclitaxel. Furthermore, since these authors specifically included paclitaxel and dasatinib, it would be particularly useful to understand overlaps in the efficacy of this combination in the present work relative to the previously described efforts (reference 20).
  9. The response to 11 monotherapies and 12 combination therapies, were tested in 12 AGCT patient-derived cell lines and an AGCT cell line (KGN) relative to breast cancer cell line MCF-7 and neuroblastoma cell line SH-SY5Y. The authors suggest that all cell lines showed synergistic growth inhibition by combination treatment with carboplatin, paclitaxel and alpelisib that are below the maximum achievable concentration in patients (IC50<Cmax). While the authors cite Cmax/Css from clinical trials, we are not provided comparisons of the concentrations achieved in vitro with the test substances. The methods (Table 2 and lines 385-386) describe the dissolution of compounds in 1% DMSO, which appears to be different from standard practice of 10mM solutions in 100% DMSO diluted to 1% DMSO to maintain solubility of typically hydrophobic small molecules. Unlike chemotherapeutic agents tested which appear to be highly water soluble, the kinase inhibitors and steroidogenic pathway inhibitors listed in table 2 are not expected to have uniform solubility in 1% DMSO at the concentrations dissolved. Kinetic solubility is easily measured with kits from commercial suppliers (e.g. Millipore) and will provide experimental confirmation of the achieved concentration to correctly associate cell culture concentrations with Css or Cmax concentrations.
  10. Carboplatin was dissolved in 0.01% Tween-20. This detergent is known to cause disruption of aggregate forms of many compounds, and since carboplatin is used with many of the combination therapies in this report, it is unclear if there is a vehicle effect of Tween-20 on the efficacy of kinase inhibitors or anti-hormones on the responses in AGCTs. Experimental kinetic solubility determinations will confirm the concentration of compounds in the combination conditions.
  11. In the context of kinetic solubility, the authors state that the 2-drug combination of everolimus+apelisib was only effective at concentrations that exceeded Cmax/Css. It would be important to confirm the exposure to drug substance in vitro in order to translate these in vitro results into clinical trial actions. Again in the context of solubility, it is important to know whether the Tween 20 associated with carboplatin increased solubility of test substances. We are not provided results with carboplatin alone to evaluate this possibility in vitro.
  12. Methods or supplementary information should outline the actual concentrations of inhibitors tested to understand what data points were used to generate the computer-smoothed curves shown in figures 2-5. No supplementary materials were found online. We also are not apprised of variability in the responses at any time point, nor are the standard error values or standard deviations provided for IC50 determinations. No statistically-determined P-values are provided (e.g. P<0.05) to reflect statistical differences in responses of various combinations.
  13. The authors rationale for selecting ulipristal is absent, while recent articles support progesterone receptor involvement in AGCTs (https://www.gynecologiconcology-online.net/article/S0090-8258(19)30057-5/fulltext). This reference suggests that high PR composite score (≥9) was associated with both decreased recurrence-free and overall survival in patients with GCT while ER expression was not associated with survival outcomes.
  14. More information about the purpose of FoxL2 analysis for purpose of confirming genetic stability of the tumor cells in vitro would be helpful in the introduction. Additionally if there are mechanistic rationale for measuring FoxL2, these should be more carefully explained in Introduction. Particularly relevant for GCPA113T1.II response where only 2 of 4 pharmacologies are represented (compared with other cell lines in Figure 4).
  15. Scientific Soundness: Many missing pieces of evidence to take this in vitro experiment and convert this into a clinical trial action. It would be particularly useful for the anti-hormonal agents used (tamoxifen, letrozole, fulvestrant, ulipristal, anastrozole) and the kinase inhibitors used (everlimus, apelisib, dasatanib, 6-THIO-2dG) to have cellular markers of the response to confirm that the survival of cells is matched with an expected signaling response of the inhibitor. It is common practice with the use of kinase inhibitors or steroid receptor antagonists to demonstrate that an accepted signaling endpoint or biomarker of response is modified. For kinase inhibitors, there has not been confirmation that decreased phosphorylation of the target substrate has been achieved. For steroid receptor antagonists, we are not provided any evidence that the antagonist either works as expected on a transcriptional endpoint (qPCR) or a secreted biomarker (ELISA) but fails to inhibit survival of the AGCTs.
  16. Reasons why this reviewer sees this to be important. The patient-derived AGCT cultures exhibit 4 dominant pharmacologic responses (Figure 4) based on the apparent non-parallelism of the inhibition curves. First, are characterized by Carboplatin +Paclitaxel +Apelisib as one set of parallel inhibition curves, Everolimus + Tamoxifen as another and dramatically different pharmacology response, and two additional sets of parallel inhibition curves (difficult to separate the different color lines in the figures). Would recommend finding a better way to display so many curves – reviewers and readers have a difficult time connecting 12 treatments with lines on graphs, especially when these are reduced to publication-sized figures.
  17. Expected to see results from positive control cell lines, breast cancer cell line MCF-7 and neuroblastoma cell line SH-SY5Y, but were not found in information provided. If they are in supplementary files, then these figures need to be provided.

SPECIFIC COMMENTS

  1. The article is well written and understandable; there are a few instances to be corrected:
  2. Compare lines 143-144 with Figure legend for Figure 2. all monotherapies the IC50 exceeded the Cmax and Css values, vs For all monotherapies, the Cmax and Css exceeded the IC50. Please correct
  3. In the introduction beginning line 64 - First line systemic treatment is currently derived from high grade epithelial ovarian cancer treatment guidelines and consists of the combination carboplatin and paclitaxel, although treatment with bleomycine, etoposide and cisplatin (BEP) is also used. A retrospective analysis investigating the efficacy of this regimen in AGCT demonstrated a partial response in 11-25% and complete response in 11-26% of patients. Please clarify if it is former (carboplatin+paclitaxel) or latter regimen (bleomycin, etoposide, cisplatin) that led to partial response and if it is same 11-25% or cumulative 11-26% that achieved

Overall Recommendation

  • Reconsider after Major Revisions:

Reviewer 3 Report

The authors are to be commended for pursuing an ambitious project to address a rare cancer badly in need of additional attention. The cooperation among multiple hospitals to accrue a meaningful number of cases of a very rare diagnosis for this complex study is emblematic of the type of team science efforts needed in the field. They take an important step towards spurring new ideas for novel therapeutic approaches for this challenging disease. Their findings of synergy of the stated triplet are compelling in its consistency among multiple patient-derived cell lines. I am not convinced, however, that these findings are sufficient to direct human trials or therapies in their current form, and I think this needs to be expressed more carefully. The title of the manuscript suggests clinical applicability, but much more information would be needed before 'personalized treatment' decisions could be influenced by this approach. Additional studies to better establish the biologic plausibility, including molecular or pharmacodynamic markers to explain the findings would be needed to contemplate clinical applicability. It remains unclear to the reader why the synergistic effect is seen, and the question of whether this is only a phenomenon relevant to this in vitro system lingers. Genomic profiling was performed and was fruitless; perhaps there are other assessments (epigenetic, etc) that could be looked at. In the modern era of targeted therapy, it is hard to accept and in vitro finding without at least some molecular corroboration. In addition, PDX models might be more compelling to at least try to replicate some of the relevant in vivo processes. These tumors can exhibit varied clinical behavior and complex tumor micro-environments, hormonal milieu, and vasculature that may affect clinical efficacy and cannot be adequately represented in vitro. In addition, some correlation between what is seen in patients and what is determined in vitro would be also extremely helpful, though understandably difficult. The authors rightly point out the limitation of using Cmax values for individual drugs, and not combos; however, this is a very important point, as combination trials of similar agents in human studies have shown unfavorable toxicity profiles.

Round 2

Reviewer 2 Report

Changes to manuscript and access to Supplemental Figures addresses previous concerns for this manuscript.

Author Response

Dear reviewer,

Thank you for your time to thoroughly review our manuscript. It has helped tremendously to improve the paper. We are very happy to learn that all points were addressed and clarified. Thanks again for you time and consideration.

With kind regards, on behalf of the research group,

Joline Roze